# Social capital as a network measure provides new insights on economic growth

**Jaime Oliver Huidobro**[1,2]*, **Alberto Antonioni**[3], **Francesca Lipari**[3], **Ignacio Tamarit**[2]

**1** Department of Mathematical Engineering, Carlos III University of Madrid, Leganés, Spain, **2** Department of Data Science, Clarity AI Europe S.L, Madrid, Spain, **3** Grupo Interdisciplinar de Sistemas Complejos (GISC), Department of Mathematics, Carlos III University of Madrid, Leganés, Spain

* jaime.oliver@clarity.ai

**Data Availability Statement:** The data underlying the results presented in the study are available from the Organisation for Economic Co-operation and Development website: - Input Output tables:

## Abstract

Unveiling the main drivers of economic growth is of paramount importance. Previous research recognizes the critical role played by the factors of production: capital and labor. However, the exact mechanisms that underpin Total Factor Productivity (TFP) are not fully understood. An increasing number of studies suggests that the creation and transmission of knowledge, factor supply and economic integration are indeed crucial. Yet, the need for a systematic and unifying framework still exists. Nowadays capital and labor are embedded into a complex network structure through global supply chains and international migration. Recent research has established a link between network centralities and different types of social capital. In this work we employ the OECD's Multi-Regional Input-Output and International Migration datasets to build the network representation for capital and labor of 63 economies during 10 years. We then examine the role of social capital measures as drivers of the TFP adopting an extended Cobb-Douglass production function and addressing potential issues such as multicollinearity, reverse causality and non-linear effects. Our results indicate that social capital in the factors of production networks can significantly drive economic outputs through TFP.

## Introduction

From the seminal work of Solow [1], economic output has been understood as a monotonically increasing function of the factors of production—land, labor and capital—plus an additional term called Total Factor Productivity (TFP) to account for any additional unknown factors. Although TFP has been found to be the key determinant of the long run growth rate (per worker) [2], its drivers remain unclear. Departing from the exogenous TFP models from Solow [1], the endogenous growth models [3, 4] addressed the exogeneity assumption by linking TFP to innovation, education, regulatory environment, and public goods and institutions. Human capital enter the scene as a determinant of positive externality that drives productivity, which in turn explains long-run growth [5]. Further contributions [6] included both human and physical capital in a composite measure that faces no decreasing returns, suggesting that continuous investment can lead to long-term growth. Also, it has been shown that differences

https://www.oecd.org/sti/ind/inter-country-input-output-tables.htm - Migration database: https://www.oecd.org/els/mig/keystat.htm.

**Funding:** J.O.H. and I.T. acknowledge Clarity AI Europe S.L. for the financial support. F.L acknowledges the Comunidad de Madrid's Programa de atracción de talento under grant agreement no. 2018-T2/SOC-11335. A.A. acknowledges the financial support of the Spanish Ministerio de Ciencia, Innovación y Universidades under the Grant No. IJC2019-040967-I.

**Competing interests:** The authors have declared that no competing interests exist.

in human capital and Research and Development in OECD countries explain cross-country differences in total factor productivity growth [7].

On the one hand we have long-term determinants like trade integration, geography [8] and institutions [9–12]. In fact, the literature identifies that friendly economic environment and policies lead to higher firm economic performance through a positive effect on TFP [13], and that financial openness leads to TFP growth [14, 15]. On the other hand there are medium-term determinants like knowledge creation, transmission and absorption, competition, factors' supply and efficient allocation. Although we recognize that the relationship between long-term and medium term determinants is strong, our paper offers a contribution to the literature of medium-term determinants, specifically on what pertains knowledge and factor supply.

Knowledge in the shape of innovation plays a key role [16] in TFP. The channels through which innovation takes place are several. Apart from knowledge creation, that occurs through domestic and foreign Research and Development investments, there are two other channels: knowledge transmission and absorption. Knowledge transfers can be imported. For instance, goods embody technological know-how. Therefore, trade connections among countries can spur such transmission of know-how; importing relatively advanced goods can potentially increase the stock of knowledge [17]. Another channel can be Foreign Direct Investments (FDI), which theoretically brings knowledge into a country [18–22]. In summary, we could argue that more open economies could be better positioned to acquire knowledge from abroad.

For what pertains factor supply, knowledge transfer may be inhibited by lack of absorptive capacity. Nonetheless, a country absorption capacity can be strengthen through the reception of skilled migrants [23–28]. Also, there are studies showing that regardless of the level of education and the skills of the inflow of foreign workers, migration *per se* can have positive effects on the productivity growth of destination countries [29–31].

Interestingly, many of these elements rely on the fact that the two main factors of production—labor and capital—flow across the globe through global supply chains and international migration networks respectively. On the trade side, traditional economics reveals that export diversification of products leads to growth [32], and especially for developing countries [33, 34]. On the migration side, studies have found that the macroeconomic and fiscal consequences of international migration are positive for the growth of OECD countries [35], and the information contained in bilateral migration stocks suggests that migration diversity has a positive impact on real GDP [23]. This effect is believed to be related to the fact that the probability to emigrate is higher for individuals with higher human capital [36]. Also, it has been found that when international asylum seekers become permanent residents, their macroeconomic impacts are positive [37]. Nonetheless, classical methods use local, first-neighbour metrics (usually Herfindahl-Hirschman Index or similar). Therefore they are not able to exploit the information at higher-order neighbours contained in the the full network structure.

Recent literature contributions capitalize on complex and network theory to reshape old fashion trade models [38, 39]. The global financial and migratory flows can be interpreted as having a complex network structure where nodes are countries and links are flows of labor and capital, and this requires sophisticated tools to be fully understood [40]. At the macro level, it has been shown that rich countries display more intense trade links and are more clustered [41]. In this trade network, node-statistic distributions and their correlation structure have remained surprisingly stable in the last 20 years [42]. At the micro level, there is evidence that node centrality on the Japanese inter-firm trading network significantly correlates with firm size and growth [43]. Also, the country-level migration stock network has been found to have a small world structure [44, 45], and another study found a network homophily effect that could be explained in terms of cultural similarities [46].

On another line of work, advances on complex network theory link network centrality measures to social capital types [47]. This concept has mainly been tested on social networks, linking social capital to information diffusion [48], innovation [49] and even personal economic prosperity [50].

In this paper we refrain ourselves from using the traditional definition of social capital of political scientist [51–59] as determinant of growth and productivity. We rather leverage the work of Bordieu [60], who defines social capital as "*the aggregate of the actual or potential resources which are linked to possession of a durable network of more or less institutionalized relationships of mutual acquaintance or recognition. . .*". In this way, we move away from the definition of social capital as the ability of people to connect with others through trust and reciprocity. Instead we focus more on the functional properties of social capital, that are the ability to acquire valuable information and favors. These in turns, are important for innovation, and hence growth.

To do this, we capitalize on the effort made by Jackson [47] to break down the definition of social capital into fundamental forms of capital and to operationalize its definition through measures of network centrality. Jackson defines capital as any "stock—other than land and labor—that can be used, or converted into something that is useful in the production or distribution of any good, service, skill, or knowledge". Under this definition, the determinants of TFP can be represented as a form of social capital, and hence their effects on growth can be captured through network centrality measures. Specifically, the determinants related to knowledge and innovation can be expression of Information capital, while the determinants associated to Factor supply can be a proper representation of favor capital.

The purpose of this work is to unify these different strands of literature under a single framework. We consider that economic transactions and migratory flows are a channel for information exchange. We further argue that a better position in these networks—higher network social capital—allows economies to access information that is crucial for innovation and, in turn, for Total Factor Productivity. We proxy two different types of social capital with two centrality measures [47]: incoming (out-coming) information capital with hubs (authorities) score, and inwards favor capital with favor centrality. In order to understand the relationship of social capital with TFP, we propose an augmented Cobb-Douglass production function [61]. In this way, we give social capital a well defined role in growth theory.

To test this model, we build the network representations of the factors of production. On the one hand, we build two representations of the capital flows network leveraging the OECD's World Multi-Regional Input Output database, one for capital and another for goods and services. On the other hand we build the labor flow network using the OECD's International Migration database. We use the percentage of flow from one country to another as the link weights. This way the proposed social capital proxies differentiate from total flows (imports, exports or migrants), but they rather focus on diversification. The whole process results on a panel data set covering 63 economies from 2005 to 2015, including several social capital indicators for each country. We then estimate the coefficients of the growth model using both OLS and Random Effects estimators.

The model might be affected by three main issues; multicollinearity, non-linear and interaction effects, and reverse causality. Multicollinearity occurs because centrality measures tends to correlates among each others. We address the issue by performing a PCA and an Elastic-Net regression. We tackle non-linear and interaction effects through a Gradient Boosting model analysis through SHAP values, pointing out that linearity might not be the best functional form for the model. Our strategy to deal with reverse causality (simultaneity bias) is to use an Arellano-Bond estimator to correctly identify the social capital effect.

The results of our analysis, which take into account the issues above, show that social capital, in the form of network centrality measures can be added as determinants of countries' TFP and as a further explanation of their difference in growth. Moreover, there are some methodological advancement that the model unveils. In a world highly interconnected in which countries have many level of interactions, i.e. financial, trading, migration network, the traditional macroeconomic models would benefit from capitalizing on the tools of complex systems, network theory and machine learning.

## Materials and methods

### Linking social capital types to TFP factors

As we pointed out in the introduction, foreign sources of knowledge, human capital and technology are linked to TFP and in turn to growth. Knowledge from abroad may flow through a variety of channels. On the one hand, knowledge on how to efficiently use the factors of production is key for productivity and, in that way, knowledge transfers among countries help develop technology and therefore drive TFP. The first channel for knowledge transfers is Foreign Direct Investments (FDI), helping knowledge spillovers from industrialised to developing countries [18–22]. The second channel is through imports of sophisticated goods and services with high technological content [17]. And the third channel is through international migration [23, 24] that works both through the reception of skilled migrants in developed economies, and through migrants' attachment to their original countries. On the other hand, availability of human capital is key to absorbing knowledge shocks, so access to foreign labor is also potentially key to TFP growth.

We propose a direct association between these factors and two types of social capital (see [47] for a review of different social capital types). The first one is information capital, a proxy for the ability to acquire valuable information and/or to spread it to others. Information capital properly measures the transfer of knowledge driven by FDI, trade and migration. The second one is favor capital, which is defined as having neighbours that are supported by a neighbour in common. Favor capital represents the hypothesis that migrants tend to migrate to cities and regions in which their compatriots have already settled, and that they tend to exploit the networks provided by their national community to find jobs.

Information capital ($I$) is related to diffusion centrality [62], which converges to eigenvector centrality in infinite iterations [63]. As both of our networks are directed, we leverage the HITS algorithm to proxy inwards ($I^{in}$) and outwards ($I^{out}$) information capital with the authorities and hubs centralities respectively [64]. On the other hand, the favor capital of node $i$ in an un-weighted network $\mathbf{g}$ as been previously proxied with favor centrality as follows [47]:

$$F_i(\mathbf{g}) = |j \in N_i(\mathbf{g}) : [\mathbf{g}^2]_{ij} > 0|. \tag{1}$$

Where $N_i(\mathbf{g})$ is the set of $i$'s neighbours—notice that the term $[\mathbf{g}^2]_{ij} > 0$ is restricting the set to neighbors of $i$ that are connected to at least another neighbor of $i$. Since the networks we deal with are weighted, we modify the definition in a natural way as follows:

$$F_j(\mathbf{g}) = \sum_{i \in N_j(\mathbf{g})} [\mathbf{g}^2]_{ij} \tag{2}$$

To better illustrate the intuition behind these measures, in Fig 1 we show the proposed social capital measures over a toy model network with link weights equal to one. For example, in figure (a) and (b) we observe that node 3 (node 1) has the highest inwards (outwards) information capital, because it is the target (source) of many links. On the other hand, in figure (c),

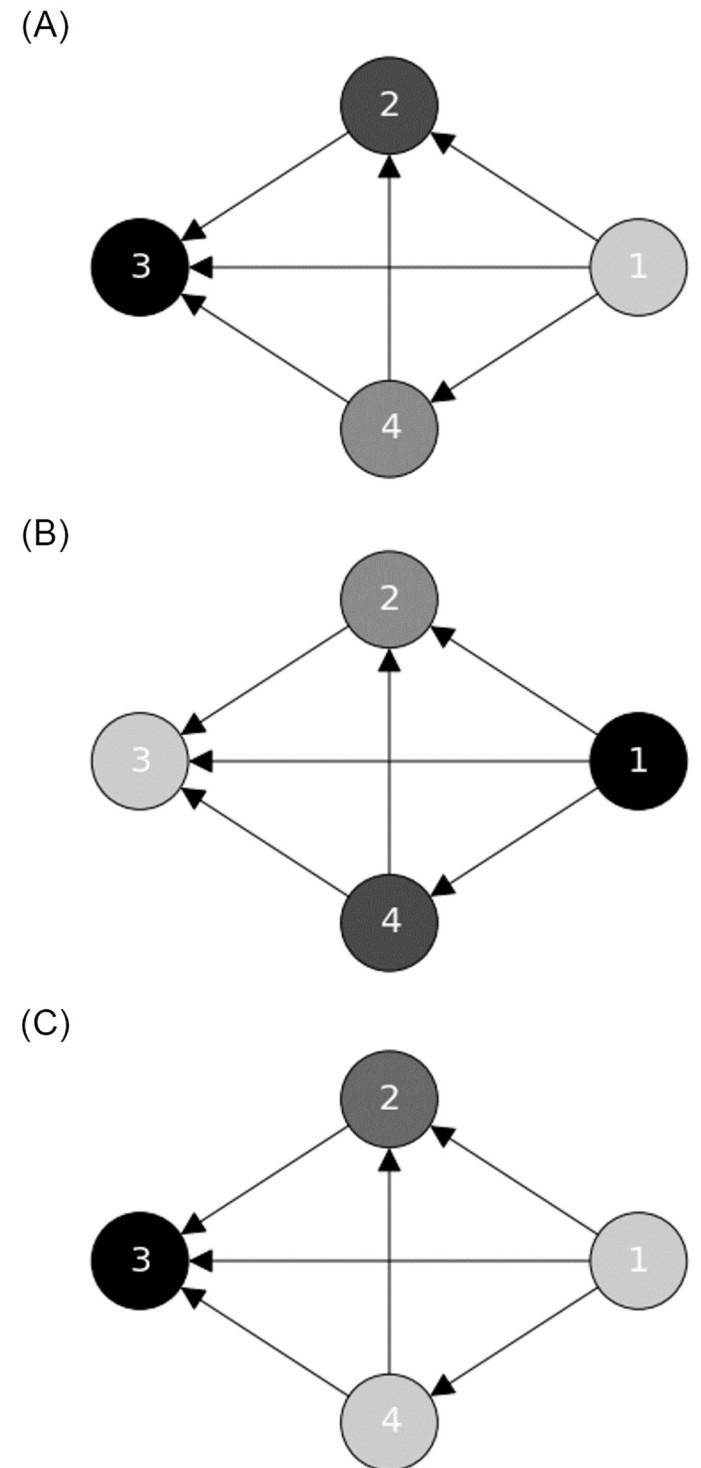

**Fig 1. Toy model of the three social capital indicators, where nodes with higher values are colored darker.** In this example all link weights are equal to one. (a) Inwards information capital. (b) Outwards information capital. (c) Inwards Favor capital.

**Table 1. Proposed relationship between different TFP growth factors and the different types of social capital in the different networks.**

| Contribution to total factor productivity | Social capital type | Proxy | |
| --- | --- | --- | --- |
| | | Network | Centrality |
| Knowledge transfer through FDI | Information capital | Financial | Hubs (out) Authorities (in) |
| Knowledge transfer through trade | Information capital | Goods and services | Hubs (out) Authorities (in) |
| Knowledge transfer through migration | Information capital | Migration | Hubs (out) Authorities (in) |
| Access to human capital supply | Favor capital | Migration | Favor centrality (Eq 2) |

we see that nodes 2 and 3 have the lowest favor capital, since they do not have any neighbours with neighbours in common.

Thus, for what concerns knowledge transfer, we link FDI with information capital on the financial network, then, we link knowledge transfers associated with importing sophisticated goods and services to information capital on the goods and services network, and knowledge transfers associated to migration with information capital on the migration network. For what concerns the human capital supply, we link this factor to favor capital in the migration network, understanding it as the belonging to country partnerships of free movement of people. The proposed links between the types of social capital and drivers of TFP are summarised in Table 1.

### Social capital in economic growth model

In order to understand the relationship of the social capital network indicators with TFP, we propose a growth model that includes the network representations of the factors of production. Macroeconomic theory generally describes a country's output through the aggregate production function [1, 3] for which one widely used functional form is the so called Cobb-Douglas function [61]:

$$Q = A \cdot K^{\alpha} \cdot L^{\beta}, \qquad (3)$$

where $Q$ represents total production, $A$ stands for the Total Factor Productivity, $K$ is capital and $L$ is labor. We propose an augmented Cobb–Douglas production function including the human and financial social capitals:

$$Q = \bar{A} \cdot K^{\alpha} \cdot L^{\beta} \cdot S(K)^{\kappa} \cdot S(L)^{\lambda}, \qquad (4)$$

where $S(x)$ stands for the social capital of the factor of production $x$. Notice that the key difference with respect to Eq 3 is that we explicitly factor out the social capital contributions from the TFP as follows:

$$A = \bar{A} \cdot S(K)^{\kappa} \cdot S(L)^{\lambda} \qquad (5)$$

### Data

In general, we interpret global trans-national interactions (both financial and migratory) as a network ($\mathcal{G}$), with $n$ countries (nodes) indexed by $i \in \{1, \ldots, n\}$. This graph is described by its adjacency matrix $\mathbf{g} \in [0, 1]^{n \times n}$, where the $g_{ij} > 0$ represents the weight of the interaction between $i$ and $j$. Since these are directed graphs, $\mathbf{g}$ is not necessarily symmetrical for any of them.

There is a growing body of literature interpreting the global financial flows as a complex network [65, 66]. Although interpreted in a different way, the adjacency matrix of the financial network has been thoroughly studied in the field of Input-Output economics [67] under the name of technical coefficient matrix, and thus there are many open data-sources providing this information. In particular, we used OECD's World Multi-Regional Input Output database to proxy the amount of trade between pairs of countries. On the one hand, we extracted the adjacency matrix of the financial network ($\mathcal{G}_F$), where the directed link weight from country $i$ to country $j$ represent the percentage of $i$'s economic output (measured in dollars) that is paid to country $j$ in exchange of goods and services exported. On the other hand, we built the goods and services network ($\mathcal{G}_G$) by weighting the links with the proportion of the total production of goods and services that a country $i$ exports to country $j$.

We build the migration network's ($\mathcal{G}_M$) adjacency matrix by leveraging the OECD's International Migration Database. This database contains information for the yearly number of people migrating from country $i$ to country $j$. Thus, we defined the weights of $\mathcal{G}_M$ as the yearly number of migrants going from country $i$ to country $j$, relative to the working population of country $i$.

In order to estimate the coefficients in the model 4, we leverage several economic indicators. Economic output is modeled with GDP (in current US dollars) provided by the World Bank, capital is modeled as Gross Fixed Capital Formation (in current US dollars) provided by the World Bank and labor as total working population (in millions) provided by the OECD. The result is a panel data set covering 63 economies and 10 years. In Fig 2 we show the distributions of the different variables as well as their pairwise Spearman correlations and $R^2$ coefficients of a linear regression model with intercept.

## Estimation methodology

We model the relationship of social capital with GDP of country $i$ at time $t$ in a linear fashion by taking logs in Eq 4:

$$\log(GDP_{it}) = A + \alpha \cdot \log(K_{it}) + \beta \cdot \log(L_{it}) +$$
$$\xi_{\mathcal{M}} \cdot \log(F_{it\mathcal{M}}) + \sum_{n \in \mathcal{N}}(\mu_n \cdot \log(I_{itn}^{in}) + v_n \cdot \log(I_{itn}^{out})) \tag{6}$$

where $A$ is the intercept, $K_{it}$ is the gross capital formation, $L_{it}$ is the total working population, $\mathcal{N}$ is the set of networks $\{\mathcal{G}_F, \mathcal{G}_G, \mathcal{G}_M\}$, $F_{itn}$ is the inwards favor capital, and $I_{itn}^{in}$ and $I_{itn}^{out}$ are the in and out information capitals respectively.

We present four specifications of the model. First, through Ordinary Least Squares (OLS), we estimate the original model coefficients which considers as regressors the original three inputs (i.e. Total factor productivity, Capital and Labour). Second, we estimate the augmented version of the first specification by adding the favor capital and the information capital. To account for unobserved entity and time effects, we leverage a Random Effects (RE) estimator including both country and year effects. We use RE for both the simple (third specification) and augmented (fourth specification) set of regressors. The choice of a RE model is sustained by the Hausman test. Additionally, the interdependence of our observations (same country appearing several years) could lead to heteroskedasticity in the residuals, and thus to underestimation of the estimated standard errors. We use heteroskedasticity and autocorrelation consistent (HAC) errors in our estimation to tackle this issue.

The results of the pooled-OLS and Random Effects estimations could suffer from multicollinearity, unaccounted non-linear and interaction effects, and simultaneity and endogeneity of the estimates. For what concerns multicollinearity, the issues is raised by the fact that centrality

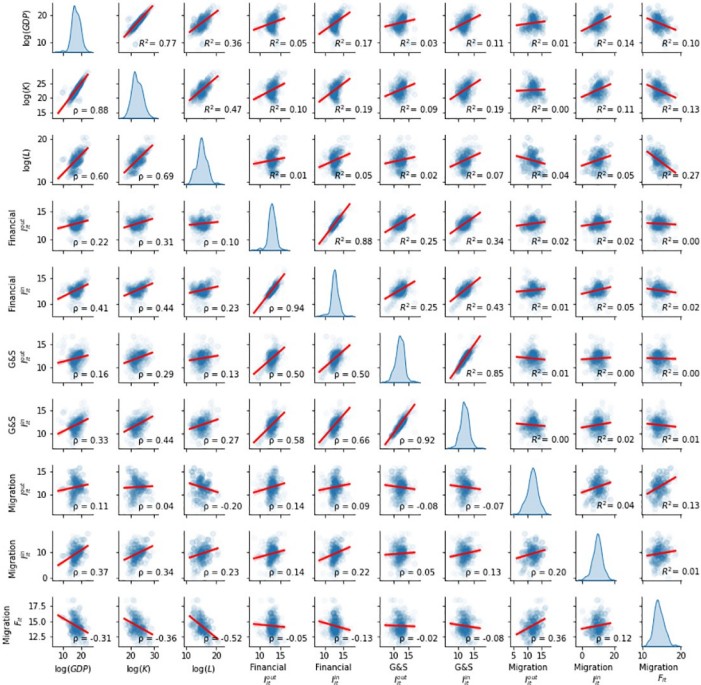

**Fig 2. Pairwise distribution matrix for economic output (log($GDP$)), capital (log($K$)), labor (log($L$)) and the developed social capital indicators: Inwards/outwards information capital ($I^{in}/I^{out}$), and favour capital $F$ for the financial, goods and services, and migration networks.** Each observation corresponds to one country and year. Spearman correlations ($\rho$) are shown in the lower triangular matrix, while the $R^2$ of a linear regression model with intercept is shown in the upper triangular matrix.

measures—and hence social capital indicators—tend to correlate [68]. Second, we haven't included non-linear nor interaction terms, so we are implicitly assuming that the effects of social capital are constant across all ranges, and are independent of the values of the rest of the variables. Finally, higher social capital enhances productivity, however higher GDP could attract trade and migration and therefore lead to higher social capital. This fact potentially introduces a simultaneity issue. We also anticipate two possible sources of endogeneity. On the one hand, growing economies most probably demonstrate higher TFP growth and attract more migrants due to a higher demand for labor. If so, this would lead to an upward bias in fixed effects estimates. On the other hand, declining sectors most probably demonstrate lower TFP; however, they might have a relatively higher presence of foreign labor force, as migrants might be more willing to accept relatively low salaries in these sectors than natives. In this case, the fixed effects estimates would be biased downwards. Therefore, to verify the validity of the obtained results we use different strategies to address the issues. We use PCA and Lasso regularisation for addressing multicollinearity, a Gradient Boosting Trees algorithm for non-linear and interaction effects, and an Arellano-Bond estimator for the identification problem.

## Results

In Table 2 we show the model estimates leveraging both pooled-ols and random effects estimators (we reject the Fixed Effects estimator through a Hausman test at the 1% significance level), with and without the social capital variables. The adjusted $R^2$ of the extended models indicates a better goodness of the fit with respect to the base models. The biggest changes happen for the $R^2_{within}$, indicating that the social capital variables mostly explain variance associated

**Table 2. Regression results for the model specification in Eq 6. p-value notation is *** , ** and * for significance at the 1%, 5% and 10% levels respectively, and standard errors are shown in parenthesis.** For each model we show number of observations N, $R^2_{overall}$, $R^2_{within}$, $R^2_{between}$, $R^2_{adjusted}$ and F-statistic.

| | Base Model | Ext. Model | Base Model RE | Ext. Model RE |
|---|---|---|---|---|
| Dep. Variable | $\log(GDP)$ | $\log(GDP)$ | $\log(GDP)$ | $\log(GDP)$ |
| Estimator | PooledOLS | PooledOLS | RandomEffects | RandomEffects |
| No. Observations | 693 | 693 | 693 | 693 |
| $R^2_{overall}$ | 0.9815 | 0.9838 | 0.9644 | 0.9767 |
| $R^2_{within}$ | 0.5791 | 0.6639 | 0.7201 | 0.7576 |
| $R^2_{between}$ | 0.9891 | 0.9899 | 0.9690 | 0.9809 |
| $R^2_{adjusted}$ | 0.9815 | 0.9838 | 0.8894 | 0.9043 |
| F-statistic | 1826.0 | 5185.9 | 2775.5 | 807.98 |
| P-value (F-stat) | 0.0000 | 0.0000 | 0.0000 | 0.0000 |
| A | -11.760*** | -9.0196*** | -9.0918*** | -8.1258*** |
| | (0.1301) | (0.4912) | (0.6871) | (1.3273) |
| $\log(K)$ | 0.9888*** | 0.8986*** | 0.7692*** | 0.7426*** |
| | (0.0041) | (0.0103) | (0.0353) | (0.0599) |
| $\log(L)$ | -0.0239*** | 0.0325*** | 0.1518*** | 0.1761*** |
| | (0.0079) | (0.0065) | (0.0263) | (0.0496) |
| $\log(I^{in}_{\mathcal{F}})$ | | -0.4204*** | | -0.6958*** |
| | | (0.0457) | | (0.1760) |
| $\log(I^{out}_{\mathcal{F}})$ | | 0.3054*** | | 0.4503*** |
| | | (0.0360) | | (0.1344) |
| $\log(I^{in}_{\mathcal{G}})$ | | 0.0094 | | 0.3131*** |
| | | (0.0530) | | (0.1102) |
| $\log(I^{out}_{\mathcal{G}})$ | | 0.0026 | | -0.1615 |
| | | (0.0688) | | (0.1252) |
| $\log(I^{in}_{\mathcal{M}})$ | | 0.0016 | | 0.0446* |
| | | (0.0095) | | (0.0248) |
| $\log(F_{\mathcal{M}})$ | | 0.0164*** | | 0.0295*** |
| | | (0.0023) | | (0.0054) |

to changes in productivity of countries across years, rather than differences in productivity across different countries. We show different $R^2$ coefficients; The standard $R^2$ is just $R^2_{overall} = Corr^2[\hat{y}_{it}, y_{it}]$. We can then focus on within-country variance through $R^2_{within} = Corr^2[\hat{y}_{it} - \hat{y}_i, y_{it} - \bar{y}_i]$, or between-country variance through $R^2_{between} = Corr^2[\hat{y}_i, \bar{y}_i]$, where $\hat{y}_*$ stands for model estimates and $\bar{y}_i = \sum_t^T y_{it}/T$. Also, we show $R^2_{adjusted} = 1 - (1 - R^2_{overall})(N - 1)/(N - p - 1)$ to adjust for additional variables in the models (see [69] for more detail). As previously mentioned, heteroskedasticity of the residuals was controlled by using HAC standard errors. Moreover, we show that the errors are homoskedastic across countries (S1 Fig).

We find all the social capital proxies but two to be statistically significant (at the 10% level), and out of those only one has a negative effect. This is consistent with the clear uni-variate relationships between the social capital indicators and $\log(GDP)$ (Fig 2). We further observe that $\alpha$ and $\beta$ undergo little changes between the base and extended models, while the main change is absorbed by the constant term (TFP). These results provide evidence for supporting the hypothesis that social capital factors drive GDP through productivity.

As we previously mentioned, there could be different issues in these estimates. We indeed find a multicollinearity issue that is confirmed by the correlations in Fig 2, but also by high

Variance Inflation Factors, i.e. a value that quantifies the severity of multicollinearity in an ordinary least squares regression analysis. To calculate the VIF of every feature, we regress it against all other features and compute $VIF_i = 1/(1 - R_i^2)$. The minimum is $VIF = 15.8$ for inwards information capital in the migratory network, and the maximum is $VIF = 2915.7$ for the outwards information capital in the financial network. Second, we do not capture either non-linear nor interaction terms. These could be of special relevance given the complex nature of the data in hand. And last, our model specification could be prune to suffer from simultaneity bias due to a reverse causality. In the next sections we perform robustness checks in order to tackle these issues.

## Robustness checks

**Multicollinearity.** We have previously shown that the correlation between the different social capital proxies is high (Fig 2), so the estimates shown in Table 2 could suffer from multicollinearity. To tackle this issue we make use of two different techniques: Principal Component Analysis (PCA) [70] and regularisation via an Elastic-Net model [71].

Through PCA we project the 6 social capital indicators into a lower-dimension and orthogonal set of 4 variables accounting for 98% of the variance in the data. In a similar way as previously done, we model $log(GDP)$ as a linear function of capital, labor and the PCA variables in the following way:

$$\log(GDP_{it}) = A + \alpha \cdot \log(K_{it}) + \beta \cdot \log(L_{it}) + \sum_{c=1}^{4} \kappa_c \cdot PCA_{it}^c \tag{7}$$

where $PCA_{it}^c$ is the value for PCA component number $c$, in country $i$ at time $t$. Following the same rationale as before, we leverage a Random Effects (RE) estimator with both country and year effects, and we use heteroskedasticity and auto-correlation consistent (HAC) errors in our estimation. We show the coefficients of the components on the different social capital indicators in Fig 3, as well as the estimates for their corresponding effects ($\kappa_c$). We find that all but the first component have significant effects. And although the coefficients are negative, these convert into a positive effect on GDP when the negative components are taken into account.

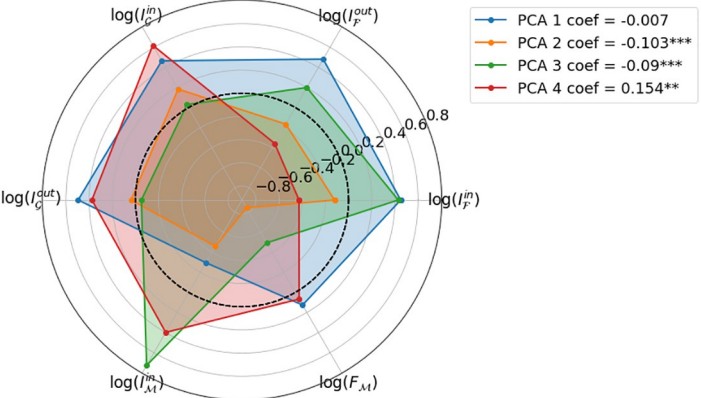

**Fig 3. Composition of the first 4 PCA components accounting for 98% of the variance in the data.** We show the estimated effects of this variables on GDP, which are mostly statistically significant.

On the other hand, we also use an Elastic-Net regression [71] to solve the multicollinearity problem by combining $L^1$ and $L^2$ regularisations [72–74] in the following way:

$$\hat{\boldsymbol{\beta}} = \arg \min_{\boldsymbol{\beta}} L(\lambda_1, \lambda_1, \boldsymbol{\beta}) \tag{8}$$

$$L(\lambda_1, \lambda_1, \boldsymbol{\beta}) = |\boldsymbol{y} - X\boldsymbol{\beta}|^2 + \lambda_1 |\boldsymbol{\beta}| + \lambda_2 |\boldsymbol{\beta}|^2 \tag{9}$$

where $\boldsymbol{y}$ is the target variable, $X$ is the matrix of regressors, $\boldsymbol{\beta}$ are the regression coefficients and $\lambda_1$ and $\lambda_2$ are the regularization hyper-parameters. The idea behind these regularisation techniques is to impose a penalty on the regression coefficients in order to provide a more sparse and parsimonious model. Through a grid-search optimization we tune the hyper-parameters to $\lambda_1 = 0.01$ and $\lambda_2 = 0.09$, making use of a 5-fold cross validation training approach to protect the results from over-fitting. Also, in order to include the country and time effects we include dummy variables for them. Finally we bootstrap our estimation in order to ensure statistical significance of the results. The resulting model scores $R^2 = 0.12$ in a pre-selected test dataset containing one third of the observations. The coefficient estimates for the social capital indicators are all positive–except for out-information capital in the migration network (Fig 4). This negative coefficient could be linked to human capital depletion. Finally, the country fixed effects show excess economic output for the US, China and Japan (S2 Fig). In summary, when we adjust for multicollinearity effects, we find evidence of statistically significant and (mostly) positive effects of social capital variables on GDP.

**Non-linear and interaction effects.** In all the previous experiments we have assumed an augmented Cobb-Douglass production function for describing economic output. However, it is well known that other production functions could apply, specially if there are non-linear and/or interaction effects. To test this hypothesis we make use of a Stochastic Gradient Boosting regressor [75, 76]. In a parallel fashion as before, we estimate $\log(GDP_{it})$ as a function of $\log(K_{it})$, $\log(L_{it})$, all the social capital indicators, and year and country dummies. We then train the model performing a grid-search optimisation for the following hyper-parameters: maximum tree depth, boosting learning rate, number of trees, and feature and observation subsampling rates. We control overfitting through 5-fold cross validation and obtain a model

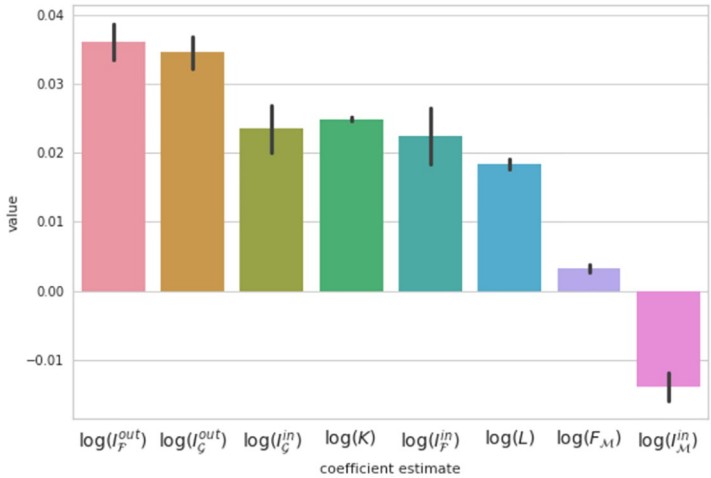

**Fig 4. Coefficient estimates for model 4 leveraging an Elastic-Net regressor to reduce the multicollinearity effect.**
Bootstrap errors are shown in black, demonstrating significance of the effects. All social capital indicators have positive effects but out-information capital in the migration network.

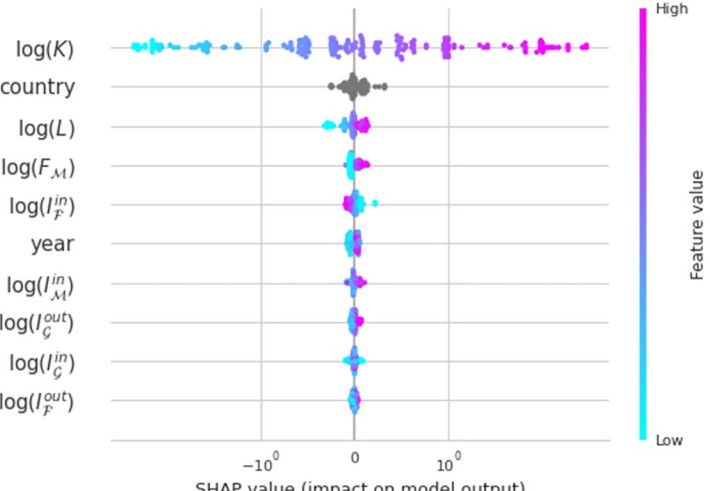

**Fig 5. SHAP values for the different features in the Gradient Boosting Regressor.** We observe that capital and country effects are the most important features, however the rest of them also contribute to the model estimates.

performance of $R^2 = 0.994$ in a pre-selected test dataset containing one third of the observations (see S3 Fig).

A well know issue with machine learning models is the lack of interpretability. Nonetheless, we leverage SHAP values [77] to analyse the feature importance in the model (Fig 5). We observe that capital significantly drives the predictions, but the rest of variables also make their contributions. Interaction effects between capital and the social capital indicators are also visible (see S4 Fig), indicating the existence of complex interaction effects between the different variables. All these results point to a more complex production function than Cobb-Douglass, containing non-linear and interaction effects.

**Reverse causality.**   An additional potential issue with the previous estimates is the reverse causality or simultaneity bias; higher social capital enhances productivity, however higher GDP might also attract trade and migration and therefore lead to higher social capital. We leverage an Arellano-Bond estimator [78] to exploit the panel structure of the data and solve the identification issue [79]. More specifically, we include the lagged GDP as an explanatory variable in model 6, so that we get:

$$\log(GDP_{it}) = \gamma \cdot \log(GDP_{it-1}) + \sum_i \eta_{it} \cdot X_{it} \tag{10}$$

where $X_{it}$ are all the previously included explanatory variables. The Arellano-Bond estimates are obtained by estimating the model in differences, and using higher-order lags of the target variable in levels as instrumental variables.

Estimation results are shown in Table 3. We observe that the estimates are consistent with the Random Effects model, albeit loosing the significance of some of the regressors. This results indicate that, although lacking statistical power for some of the variables, the contributions of social capital to GDP are robust to the reverse causality issue.

## Conclusion

The methodological contribution of this work is twofold: first, leveraging recent advances in network theory we link the social capital expressions on the global trade and migration networks to the Total Factor Productivity (TFP) of countries. And second, through an augmented

**Table 3. Results for the Arellano-Bond estimates of model 10. p-value notation is *** , ** and * for significance at the 1%, 5% and 10% levels respectively.** Adjusted $R^2$ = 0.102, and $F - stat$ = 35.671. We observe that the social capital effects maintain the same signs and similar magnitudes as in the random effects estimates (in Table 2), however some of the effects loose their statistical significance.

|  | Parameter | Std. Err. |
|---|---|---|
| $\log(I_{\mathcal{F}}^{in})$ | -0.7255*** | 0.2526 |
| $\log(I_{\mathcal{F}}^{out})$ | 0.8433*** | 0.2663 |
| $\log(I_{\mathcal{G}}^{in})$ | 0.2218 | 0.2638 |
| $\log(I_{\mathcal{G}}^{out})$ | -0.1130 | 0.2726 |
| $\log(I_{\mathcal{M}}^{in})$ | 0.0369 | 0.0301 |
| $\log(F_{\mathcal{M}})$ | 0.0129 | 0.0094 |
| intercept | 0.0296*** | 0.0058 |
| $\log(GDP_{t-1})$ | 0.2374*** | 0.0639 |

Cobb-Douglass production function we identify positive and significant effects of social capital on GDP.

In today's global economy, the two factors of production—capital and labor—travel across the globe via the mobility networks of trade and migration. And recent literature allows us to proxy the social capital of countries on these networks via distinct node centrality measures. This way we derive intuitive indicators of the topological importance of countries in the different factors of production networks, and use them to proxy a country's topological social capital.

We identify two channels for social capital to influence a country's GDP through Total Factor Productivity. On the one hand, information capital in the financial, goods and services, and migration networks is linked to knowledge transfers through FDI, trade, and migration respectively. On the other hand, inwards favor capital on the migration network is linked to access to human capital supply.

The contributions of the social capital factors to GDP are linearly modeled through an extended Cobb-Douglass production function (Eq 4). To test the model, we build two representations of the trade network—one for funds and the other for goods and services—and one representation of the migration network. To do it, we leverage the OECD's World Multi-Regional Input-Output database and the OECD's International Migration Database respectively. The result is a panel dataset with seven different social capital indicators for 63 countries across 10 years.

Leveraging OLS and Random Effects estimators we find positive significant effects of social capital on economic performance. In both cases, the fit to the extended Cobb-Douglass model is enhanced by the inclusion of the social capital indicators (especially on the within-country variance). These results are consistent with the observed positive Spearman correlations between $\log(GDP)$, $\log(K)$ and $\log(L)$ with the network centrality variables.

Nonetheless, we identify three possible issues in the effect estimation: multicollinearity, non-linear and interaction effects, and reverse causality. Therefore, to ensure the robustness of our first findings we implement a battery of robustness checks. We first tackle multicollinearity by applying PCA over the social capital indicators. We extract 4 components accounting for 98% of the data variance, and we find significant effects when regressing the components against $\log(GDP)$ controlling for capital and labor. The results are directionally aligned with our hypothesis, although the ability to interpret them is lowered by the use of this technique. Therefore, we take a second approach to reducing multicollinearity: Elastic-Net regression. The results indicate a positive effect for all the social capital indicators, except for a negative effect in human hubs that can be linked to human capital depletion.

The second conceivable issue with our estimates is the non-linear and interaction effects. To account for this, we leverage a non-linear machine learning model—Gradient Boosting Regressor. This technique obtains a much better fit to the data, confirming the existence of such effects. Through the use of SHAP values we confirm the hypothesized contribution of the social capital indicators to the model. Our results also point to potential interaction effects between the social capital variables and the factors of production.

The final issue we cover is the possible existence of a reverse causality channel between economic growth and social capital. To correctly identify the social capital effect, we leverage an Arellano-Bond estimator. Our results once again with the first model (all coefficients conserve the sign). However, we lack the statistical power to identify the model completely, and therefore some of the indicators lose their statistical significance.

In future work, we would like to control for the skills of the migrants, however at the time of writing we have no access to such dataset. Also, these results could be extended by employing a gravity model [80] of trade and migration as an instrumental variable approach to ensure the causal interpretation of the results.

Our results indicate that the presented indicators are very rich signals for policy-making. Social capital (as a network topological measure) is a latent variable that is difficult to quantify, yet it contributes to productivity and growth. First, we conclude that information social capital has a positive effect on a country's economy, which remarks the importance of knowledge transfers occurring through trade and migration. Second, considering social capital in its favor function, we confirm the positive impact of access to human capital on economic prosperity.

Finally, we hope that this work contributes to enlarging the discussion in the intersection of complex systems, economic and network theory, as they are all needed to understand the patterns of mobility and the factors of production.

## Supporting information

**S1 Fig. Residuals of Random Effects model by country.** We observe homoskedasticity of the residuals across different countries.
(TIF)

**S2 Fig. Country fixed effects.** Elasic-Net estimates World Map containing the country level fixed effects coefficients estimated with the Elastic-Net regressor. We observe excess economic output for the US, China and Japan.
(TIF)

**S3 Fig. Gradient Boosting Regressor residuals.** Gradient Boosting Regressor model residuals for both the training and test sets. We observe very high model performances in both, as well as homoskedasticity of the residuals.
(TIF)

**S4 Fig. SHAP interaction effects.** SHAP interaction effects for the different features in the Gradient Boosting Regressor. We detect strong interaction patterns between the social capital, and capital and labor respectively. This result provides evidence for the existence of interaction effects between the model variables.
(TIF)

## Acknowledgments

All the authors acknowledge Marta Rivera Alba for her support at the early stages of the project, as well as Marco Pangallo for his useful feedback. Also, the authors acknowledge the 10th

International Conference on Complex Networks & Their Applications for hosting the presentation of the proceedings paper.

## Author Contributions

**Conceptualization:** Jaime Oliver Huidobro, Alberto Antonioni, Francesca Lipari, Ignacio Tamarit.

**Data curation:** Jaime Oliver Huidobro.

**Formal analysis:** Jaime Oliver Huidobro.

**Investigation:** Jaime Oliver Huidobro, Alberto Antonioni, Francesca Lipari, Ignacio Tamarit.

**Methodology:** Jaime Oliver Huidobro, Alberto Antonioni, Francesca Lipari.

**Supervision:** Francesca Lipari, Ignacio Tamarit.

**Writing – original draft:** Jaime Oliver Huidobro, Alberto Antonioni, Francesca Lipari, Ignacio Tamarit.

**Writing – review & editing:** Jaime Oliver Huidobro, Alberto Antonioni, Francesca Lipari, Ignacio Tamarit.

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
