## [Decision Letter · Decision Letter 0]

11 May 2022

PONE-D-22-07302Social capital as a network measure provides new insights on economic growthPLOS ONE

Dear Dr. Oliver Huidobro,

Thank you for submitting your manuscript to PLOS ONE. After careful consideration, we feel that it has merit but does not fully meet PLOS ONE’s publication criteria as it currently stands. Therefore, we invite you to submit a revised version of the manuscript that addresses the points raised during the review process.

We look forward to receiving your revised manuscript.

Kind regards,

Hocine Cherifi

Academic Editor

PLOS ONE

Journal Requirements:

3. We note that Figure S1 in your submission contain map images which may be copyrighted. All PLOS content is published under the Creative Commons Attribution License (CC BY 4.0), which means that the manuscript, images, and Supporting Information files will be freely available online, and any third party is permitted to access, download, copy, distribute, and use these materials in any way, even commercially, with proper attribution. For these reasons, we cannot publish previously copyrighted maps or satellite images created using proprietary data, such as Google software (Google Maps, Street View, and Earth). For more information, see our copyright guidelines: http://journals.plos.org/plosone/s/licenses-and-copyright.

a. You may seek permission from the original copyright holder of Figure S1 to publish the content specifically under the CC BY 4.0 license.  

Reviewers' comments:

Reviewer's Responses to Questions

**Comments to the Author**

1. Is the manuscript technically sound, and do the data support the conclusions?

Reviewer #1: No

Reviewer #2: Yes

2. Has the statistical analysis been performed appropriately and rigorously? 

Reviewer #1: No

Reviewer #2: Yes

3. Have the authors made all data underlying the findings in their manuscript fully available?

Reviewer #1: No

Reviewer #2: Yes

4. Is the manuscript presented in an intelligible fashion and written in standard English?

Reviewer #1: Yes

Reviewer #2: Yes

5. Review Comments to the Author

Reviewer #1: Thank you to the authors for this interesting paper examining network drivers of economic growth.

1. The authors are testing how economic connectivity (a la trade and migration networks) affects economic growth, which is a thoughtful and interesting hypothesis. Their measures seem plausibly related to growth (although as I mention below, I believe they are measuring economic connectivity, not social capital, which should be corrected before publication).

2. The authors measure economic connectivity/centrality in an economic network of OECD countries using financial trade flow data. They call these centrality measures 'social capital' - which is most certainly is not. Social capital refers to grassroots level ties that connect people, fostering trust and reciprocity (eg. Putnam, Coleman, Lin, any of the core literature on social capital). However, the authors have not measured social capital; they have measured economic connectivity, or perhaps economic centrality among countries. While deserving of inquiry, this paper does not actually test hypotheses about social capital but rather national level economic networks, and should be retitled and adapted accordingly.

3. It's a very sophisticated study, but I'm having a little trouble seeing the value added right now. We would naturally expect countries with more GDP probably end up making more (1) outward foreign direct investment elsewhere, because they have the money to do so. Countries with greater GDP (dependent variable) will also see more (2) inward trade and more (3) in-migration. These are basically three mutual proxies for the same concept, so it's a little unclear what the puzzle is here. Put more positively, could the authors please justify more clearly what gap this research is trying to address?

4. Their modeling strategy needs to address the inter-dependence of observations problem which occurs when using network centrality as a predictor/outcome (in this case, as predictors). Their observations' residuals will be correlated (heteroskedasticity) because the country-year observations are literally connected to one another (inter-dependent observations); they will often share similar predictor values. Heteroskedasticity compromises the validity of beta coefficient standard errors. There are a few solutions for this. The authors could (a) permute the networks themselves to calculate the median expected centrality at random, and then subtract that from the original centrality measure. This would give you a residual centrality measure that is corrected for network structure, and could be appropriately regressed as-is. This is admittedly kind of tricky and requires some coding, so I might suggest instead you try an alternative: (b) use statistical simulation (eg. CLARIFY in Stata, Zelig in R, etc.) to simulate the effects of your key independent variables on expected change in log GDP from a multivariate normal distribution. This uses your model equation as is, but does not rely on beta coefficients' standard errors, and so is robust to heteroskedasticity. This might be the easiest move here.

Minor notes:

5. I am not personally an expert in elastic-net regression or gradient boosting models, and so cannot comment on these methods. I will just say that a simpler, more robust way to hypothesis test given correlated predictors would be to just use predicted values/statistical simulations as discussed above, since correlated predictors and heteroskedasticity only affect beta coefficients' standard errors, not model predictions.

6. Table 2 and later figures are difficult to read, because the authors have chosen to label their predictors using Greek letters, leaving readers to flip back and forth just to read the table. Please sub in simple English labels!

7. The authors appear to have linked raw data sources for a few of their variables, but it looks like PLOS ONE wants authors to share or embargo their entire replication code/data. FYI.

Thank you for the opportunity to review this research. An exciting application of international trade and migration networks!

Reviewer #2: The document is clear, well written, and the topic is relevant and current. At the same time, the approach and techniques used for the analysis are pertinent and relatively new to the economics discipline.

6. PLOS authors have the option to publish the peer review history of their article (what does this mean?). If published, this will include your full peer review and any attached files.

Reviewer #1: No

Reviewer #2: No

---

## [Author Response · Author response to Decision Letter 0]

7 Jul 2022

### ------------------ ### Reviewer 1 ### ------------------ ### 

Thank you to the authors for this interesting paper examining network drivers of economic growth.

--------------

Comment 1: The authors are testing how economic connectivity (a la trade and migration networks) affects economic growth, which is a thoughtful and interesting hypothesis. Their measures seem plausibly related to growth (although as I mention below, I believe they are measuring economic connectivity, not social capital, which should be corrected before publication).

Answer: Thank you for the comment. We included a clarification in the draft in lines 72-79. 

As its mentioned in (Jackson 2019), social capital was defined by Pierre Bourdieu as (Bourdieu 1986) “the aggregate of the actual or potential resources which are linked to possession of a durable network of more or less

institutionalized relationships of mutual acquaintance or recognition...”. This definition revolves around interactions among parties, not around individual properties. 

--------------

Comment 2: The authors measure economic connectivity/centrality in an economic network of OECD countries using financial trade flow data. They call these centrality measures 'social capital' - which is most certainly is not. 

Social capital refers to grassroots level ties that connect people, fostering trust and reciprocity (eg. Putnam, Coleman, Lin, any of the core literature on social capital). However, the authors have not measured social capital; they have measured economic connectivity, or perhaps economic centrality among countries. 

While deserving of inquiry, this paper does not actually test hypotheses about social capital but rather national level economic networks, and should be retitled and adapted accordingly.

Answer: Thank you again for your comment.

In this work we consider that economic transactions and migratory flows are a channel for information exchange. And as explained in lines 25-34, one of the drivers of TFP is the exchange of information through economic exchanges. 

The focus of the work is not total imports and exports (economic connectivity), but rather the share of monetary and migratory flows (We clarified this matter in lines 104-107). The main point is that diversification and positioning in the global networks provides access to information that is fundamental for innovation and, in turn, for productivity. This positioning in the network is what we identify as social capital. And using network centrality measures allows us to capture complex measures of diversification. 

The functional perspective (i.e. information access) of social capital is embedded in the structural perspective. Therefore we are interested in the structural properties of the networks and the role of countries in them. 

--------------

Comment 3: It's a very sophisticated study, but I'm having a little trouble seeing the value added right now. 

We would naturally expect countries with more GDP probably end up making more 

(1) outward foreign direct investment elsewhere, because they have the money to do so. Countries with greater GDP (dependent variable) will also see more 

(2) inward trade and more 

(3) in-migration. 

These are basically three mutual proxies for the same concept, so it's a little unclear what the puzzle is here. Put more positively, could the authors please justify more clearly what gap this research is trying to address?

Answer: Thanks for the comment. 

The value added of the paper can be summarized in equation 5, where we propose that part of TFP can be explained as the social capital of countries in their network structure expression. We clarify this matter in lines 91-94

Our belief is that the mentioned concerns are already covered since: 

We are not using total imports/exports or total migration, but rather share (expressed as a percentage) by country. It's not about how much you export locally, but the centrality as a global network measure. We clarified this further in lines 104-107. 

Moreover, as the reviewer points out, it's true that the causal effect could be the other way around; more GDP could imply more economic and migratory centrality. However, we have already addressed this issue in the robustness check in line 327 with the AB estimates. 

--------------

Comment 4: Their modeling strategy needs to address the inter-dependence of observations problem which occurs when using network centrality as a predictor/outcome (in this case, as predictors). Their observations' residuals will be correlated (heteroskedasticity) because the country-year observations are literally connected to one another (inter-dependent observations); They will often share similar predictor values. 

Heteroskedasticity compromises the validity of beta coefficient standard errors. There are a few solutions for this. The authors could (a) permute the networks themselves to calculate the median expected centrality at random, and then subtract that from the original centrality measure. This would give you a residual centrality measure that is corrected for network structure, and could be appropriately regressed as-is. This is admittedly kind of tricky and requires some coding, so I might suggest instead you try an alternative

(b) use statistical simulation (eg. CLARIFY in Stata, Zelig in R, etc.) to simulate the effects of your key independent variables on expected change in log GDP from a multivariate normal distribution. This uses your model equation as is, but does not rely on beta coefficients' standard errors, and so is robust to heteroskedasticity. This might be the easiest move here

Answer: Thank you for the comment that helped us to clarify this concern.

We explicitly use Heteroskedastic Autocorrelated Consistent (HAC) standard errors in our estimates, in order to avoid such issues. We clarified this in lines 229-233. Moreover, we have added an additional figure to the appendix showing that errors are homoscedastic across countries.

I am not personally an expert in elastic-net regression or gradient boosting models, and so cannot comment on these methods. 

I will just say that a simpler, more robust way to hypothesis test given correlated predictors would be to just use predicted values/statistical simulations as discussed above, since correlated predictors and heteroskedasticity only affect beta coefficients' standard errors, not model predictions.

Answer: Thank you again for the comment. We have included a robustness check section for multicollinearity issues. Elastic Net solves this issue, so we have added 2 more references as well as clarified the explanation of this matter in lines 311-313. 

About statistical simulations for standard errors, we show bootstrapped standard errors for the elastic net coefficient estimates in Fig 4.

--------------

Comment 5: Table 2 and later figures are difficult to read, because the authors have chosen to label their predictors using Greek letters, leaving readers to flip back and forth just to read the table. Please sub in simple English labels!

Answer: Thank you for the comment. We changed the labels in the tables to make them more readable. 

--------------

Comment 6: The authors appear to have linked raw data sources for a few of their variables, but it looks like PLOS ONE wants authors to share or embargo their entire replication code/data. FYI.

Answer: Thanks for your comment, we will follow all guidelines required by the journal. 

### ------------------ ### Reviewer 2 ### ------------------ ### 

The document is clear, well written, and the topic is relevant and current. At the same time, the approach and techniques used for the analysis are pertinent and relatively new to the economics discipline.

Answer: Thank you for the revision and the support. 

### ------------------ ### Journal Comments ### ------------------ ### 

--------------

Comment 1: When submitting your revision, we need you to address these additional requirements. Please ensure that your manuscript meets PLOS ONE's style requirements, including those for file naming. Please update your submission to use the PLOS LaTeX template. 

Answer: Thank you for the feedback. We used the Latex template from Plos One’s website. We will upload all the files to reproduce the manuscript. 

--------------

Comment 2: We note that Figure S1 in your submission contain map images which may be copyrighted. All PLOS content is published under the Creative Commons Attribution License (CC BY 4.0), which means that the manuscript, images, and Supporting Information files will be freely available online, and any third party is permitted to access, download, copy, distribute, and use these materials in any way, even commercially, with proper attribution. For these reasons, we cannot publish previously copyrighted maps or satellite images created using proprietary data, such as Google software (Google Maps, Street View, and Earth). 

Answer: Thank you again for the comment. All figures were created by the authors with the specific purpose of this project. They are all therefore free of copyright.

--------------

Comment 3: Please include captions for your Supporting Information files at the end of your manuscript, and update any in-text citations to match accordingly.

Answer: Thank you again for the feedback. We added captions to the figures and updated the citations accordingly, including the new figure in the Supporting information file. 

--------------

Comment 4: We note that the grant information you provided in the ‘Funding Information’ and ‘Financial Disclosure’ sections do not match. When you resubmit, please ensure that you provide the correct grant numbers for the awards you received for your study in the ‘Funding Information’ section.

Answer: Thank you for pointing out the inconsistencies in the submission.

---

## [Decision Letter · Decision Letter 1]

3 Aug 2022

Social capital as a network measure provides new insights on economic growth

PONE-D-22-07302R1

Dear Dr. Oliver Huidobro,

We’re pleased to inform you that your manuscript has been judged scientifically suitable for publication and will be formally accepted for publication once it meets all outstanding technical requirements.

Kind regards,

Hocine Cherifi

Academic Editor

PLOS ONE

Reviewers' comments:

Reviewer's Responses to Questions

**Comments to the Author**

1. If the authors have adequately addressed your comments raised in a previous round of review and you feel that this manuscript is now acceptable for publication, you may indicate that here to bypass the “Comments to the Author” section, enter your conflict of interest statement in the “Confidential to Editor” section, and submit your "Accept" recommendation.

Reviewer #2: All comments have been addressed

2. Is the manuscript technically sound, and do the data support the conclusions?

Reviewer #2: (No Response)

3. Has the statistical analysis been performed appropriately and rigorously? 

Reviewer #2: (No Response)

4. Have the authors made all data underlying the findings in their manuscript fully available?

Reviewer #2: (No Response)

5. Is the manuscript presented in an intelligible fashion and written in standard English?

Reviewer #2: (No Response)

6. Review Comments to the Author

Reviewer #2: (No Response)

7. PLOS authors have the option to publish the peer review history of their article (what does this mean?). If published, this will include your full peer review and any attached files.

Reviewer #2: No

---

## [Editor Report · Acceptance letter]

16 Aug 2022

PONE-D-22-07302R1 

Social capital as a network measure provides new insights on economic growth 

Dear Dr. Oliver Huidobro :

I'm pleased to inform you that your manuscript has been deemed suitable for publication in PLOS ONE. Congratulations! Your manuscript is now with our production department. 

Kind regards, 

on behalf of

Professor Hocine Cherifi 

Academic Editor

PLOS ONE